# Repeated Antigen-Based Rapid Diagnostic Testing for Estimating the Coronavirus Disease 2019 Prevalence from the Perspective of the Workers’ Vulnerability before and during the Lockdown

**DOI:** 10.3390/ijerph18041638

**Published:** 2021-02-09

**Authors:** Ourania S. Kotsiou, Ioannis Pantazopoulos, Dimitrios Papagiannis, Evangelos C. Fradelos, Nikolaos Kanellopoulos, Dimitra Siachpazidou, Paraskevi Kirgou, Dimitra S. Mouliou, Athanasios Kyritsis, Georgios Kalantzis, Georgios K. D. Saharidis, Efthymios Tzounis, Konstantinos I. Gourgoulianis

**Affiliations:** 1Department of Nursing, School of Health Sciences, University of Thessaly, GAIOPOLIS, 41110 Larissa, Greece; evagelosfradelos@gmail.com; 2Department of Respiratory Medicine, Faculty of Medicine, University of Thessaly, BIOPOLIS, 41110 Larissa, Greece; pantazopoulosioannis@yahoo.com (I.P.); ndkanell@yahoo.com (N.K.); sidimi@windowslive.com (D.S.); Paraskevi.kirgou@gmail.com (P.K.); demymoole@gmail.com (D.S.M.); thanoskyrit@hotmail.com (A.K.); kgourg@med.uth.gr (K.I.G.); 3Public Health & Vaccines Lab, Department of Nursing, School of Health Sciences, University of Thessaly, GAIOPOLIS, 41110 Larissa, Greece; dpapajon@gmail.com; 4Department of Mechanical Engineering, University of Thessaly, Leoforos Athinon, 8 Pedion Areos, 38334 Volos, Greece; george.kalantzis4@gmail.com (G.K.); saharidis@gmail.com (G.K.D.S.); 5Volos Health Center, Thrakon 20, 38333 Volos, Greece; etzounis@yahoo.gr

**Keywords:** antigen rapid test, employment, job, lockdown restrictions, screening

## Abstract

Background: No previous study has investigated the SARS-CoV-2 prevalence and the changes in the proportion of positive results due to lockdown measures from the angle of workers’ vulnerability to coronavirus in Greece. Two community-based programs were implemented to evaluate the SARS-CoV-2 prevalence and investigate if the prevalence changes were significant across various occupations before and one month after lockdown. Methods: Following consent, sociodemographic, clinical, and job-related information were recorded. The VivaDiag™ SARS-CoV-2 Antigen Rapid Test was used. Positive results confirmed by a real-time Reverse Transcriptase Polymerase Chain Reaction for SARS-COV-2. Results: Positive participants were more likely to work in the catering/food sector than negative participants before the lockdown. Lockdown restrictions halved the new cases. No significant differences in the likelihood of being SARS-CoV-2 positive for different job categories were detected during lockdown. The presence of respiratory symptoms was an independent predictor for rapid antigen test positivity; however, one-third of newly diagnosed patients were asymptomatic at both time points. Conclusions: The catering/food sector was the most vulnerable to COVID-19 at the pre-lockdown evaluation. We highlight the crucial role of community-based screening with rapid antigen testing to evaluate the potential modes of community transmission and the impact of infection control strategies.

## 1. Introduction

The coronavirus disease 2019 (COVID-19) pandemic has been declared a health emergency of international concern. To slow down the waves of infection, the states have severely restricted economic activity [1]. Accordingly, responding to the ongoing novel coronavirus outbreak, Greece implemented the largest quarantines in the country’s history early with one of the strictest sets of lockdown measures in Europe [2]. During the first wave, a three-phase approach was adopted, starting on the 10th of March and finishing after a 42-day lockdown, on the 4th of May, when Greece gradually lifted restrictions on movement and restarted business activity. During the second wave, Greece implemented new infection preventive and control regulations relating to free movement and business activity from the 7th of November 2020. Primary schools and kindergartens have been closed since the 14th of November 2020, and since the 18th of November 2020, they switched to distance learning [3].

Work-related factors may be partly responsible for disproportionate COVID-19 infection among vulnerable groups [4]. The risk of contracting COVID-19 through work depends on its prevalence in the local community, the extent to which the job entails either proximity to people who could be carrying the infection or contact with material that might be contaminated by the virus and the effectiveness of transmission and protective measures. COVID-19 screening programs in population groups could help control the pandemic by monitoring the transmission dynamics and evaluating the effectiveness of infection control measures [1]. Hence, in the ongoing COVID-19 pandemic, timely and best available surveillance data are crucial to guide policy decisions. The first results from SARS-CoV-2 serosurveys in health care workers have begun to appear in the literature [4,5], but very limited research has examined other categories of essential workers in the general population. It has been documented recently that the risk of in-hospital transmission in healthcare workers is higher, and a lower risk of seropositivity in participants working in other community support services or those who were working from home in Italy [5].

Rapid antigen tests can contribute to overall COVID-19 testing capacity, offering advantages in terms of shorter turnaround times and reduced costs, and are the only workable option for massive screening campaigns in the general population [6]. However, no previous study has investigated the prevalence of SARS-CoV-2 infection using rapid antigen testing and the changes in the proportion of positive results in terms of the lockdown measures from the angle of workers’ vulnerability to COVID-19. This study evaluated the COVID-19 prevalence and investigated if the prevalence changes were significant across various job groups in one of the largest Municipalities in Central Greece, Volos.

## 2. Materials and Methods

Two passive surveillance programs, each of two days’ duration at two exceptional time points, before (5–6 November 2020) and one month after the lockdown initiation (30 November–1 December 2020), were conducted. The study was carried out in a population-based sample at multiple locations of Volos. Volos is situated midway on the Greek mainland and has an estimated population of 144,449. All the citizens of Volos were invited to participate in the study.

The study was approved by the Ethics Committee of the University Hospital of Larissa, and all subjects provided written informed consent. Following consent, demographic and job-related information, and data regarding patient contacts, previous testing, the participants’ travel history, and respiratory symptoms the last 15 days, medical and smoking history were recorded on questionnaire forms for all participants ( Appendix A). In addition, data related to supermarket visit frequency were collected during the second screening program.

The research was effectuated via the VivaDiag™ SARS-CoV-2 Antigen Rapid Test ((VivaChek Biotech, Hangzhou, China) [7]. Utilizing a lateral flow-based technology, the test kit detects the presence of the nucleocapsid protein antigen from SARS-CoV-2, providing 100% specificity, 90.90% sensitivity, 98.79% clinical accuracy, and 100% repeatability, and no cross-reaction with other viruses or other limitations [7]. According to test protocol, nasal swab sampling was preferred for participants’ tolerance, and no blood or other visual impurities were observed in the specimens [7]. COVID-19 antigen rapid tests were conducted by experienced doctors and skilled nurses. The evaluation of antigen-based rapid detection tests was performed by one experienced biochemist and one biologist. Subsequently, according to World Health Organizations’ (WHO) recommendations [8], cases with a positive rapid test or inconclusive results were confirmed by an opened real-time Reverse Transcriptase Polymerase Chain Reaction (RT-PCR) molecular diagnosis for SARS-COV-2, monitored by professional clinicians.

RT-PCR was performed using a TaqPath™ COVID-19 CE-IVD RT-PCR Kit, by the American ThermoFisher Scientific Inc., detecting the ORF1ab, S, N genetic loci of SARS-CoV-2, with a limit of detection (LoD) score of 10GCE/reaction [9]. As per guidelines, thermal cycling was performed as follows: 53 °C for 10 min for reverse transcription, 95 °C for 2 min, and then 40 cycles of 95 °C for 3 s, 60 °C for 30 s. The procedure followed the kits’ interim guidance ideal temperatures. All positive results showed positivity for 2 or 3 genetic loci apart from rapid tests’ N protein. All cases showed a direct and low Ct positivity, so the cutoff kit (Ct ≤ 37) was negligible [9]. The RT-PCR assay has analytical sensitivity and specificity of greater than 95% [9].

Statistical analyses were performed with IBM SPSS statistics 23. Quantitative variables were presented as frequencies or mean ± standard deviation (SD). Outliers were detected by the Grubbs test. Comparisons of frequencies were performed with the χ2 test. The normality of the data was assessed with the D’Agostino–Pearson Omnibus normality test where appropriate. Parametric data comparing two groups were analyzed with an unpaired *t*-test, while non-parametric data were analyzed with the Mann–Whitney U test. Parametric data comparing three or more groups were analyzed with one-way ANOVA and Tukey’s multiple comparisons test, while non-parametric were analyzed with Kruskal–Wallis test and Dunn’s multiple comparison test. Spearman’s correlation was used for correlation analysis. Multiple logistic regression was used to examine a series of predictor variables to determine those that best predict a positive SARS-CoV-2 Antigen test result.

## 3. Results

One thousand and fifty-four volunteers participated in the pre-lockdown screening program, and 462 people participated in the second two-day COVID-19 screening program, which was implemented one month after lockdown initiation. All volunteers were tested using rapid antigen testing. There were no inconclusive tests. All cases manifesting a rapid test positivity for SARS-CoV-2 were immediately resampled for a RT-PCR confirmation of SARS-COV-2.

### 3.1. Results from the Pre-Lockdown COVID-19 Screening Program

One thousand and fifty-four volunteers (536 males) with a mean age of 42 ± 19 years (min: 4, max: 89 years) participated in the pre-lockdown screening program. Eighty-eight out of one thousand and fifty-four (8%) had positive rapid antigen test results; all were confirmed by RT-PCR tests. The newly diagnosed participants were significantly younger than negative ones (36.4 ± 15.7 vs. 42.0 ± 17.8 years, *p* = 0.002) and belonged mainly to the age group of 20 to 40 years. Two-thirds of the positive participants had experienced respiratory symptoms in the last 15 days, while one-third were asymptomatic. Low-grade fever, loss of or change to smell or taste were significantly more prevalent among positive participants than in healthy individuals (67% vs. 19%, *p* < 0.001). No sex or other demographic differences were found in the prevalence of positive tests. Demographic characteristics and SARS-CoV-2 prevalence stratified by job group are shown in Table 1.

One-third of all participants were involved in the education sector, 9% of whom were positive. Among retired older adults (18% of the study group), only 4% were positive. Interestingly, among those working in the catering/food sector (5% of the study group), 35% were positive. Among those working in the health sector, 13% were positive at the time of testing. The distribution of positive and negative test results by job type are presented in Table 2.

Positive participants were less likely to be retirees (8% vs. 19%, *p* = 0.005) or civil servants (7% vs. 14%, *p* = 0.036) than negative participants. On the contrary, positive participants were more likely to work in the catering/food sector (19% vs. 3%, *p* < 0.001) than negative participants. Furthermore, one-third of the positive tests were found within the teaching and education sector (students, university students, teachers). Importantly, 20% of the newly diagnosed patients were university students. Forty-three percent (3/7), 44% (8/18), and 20% (1/5) of the newly diagnosed COVID-19 were students, university students, and teachers, respectively, were asymptomatic. Five percent of the newly detected patients worked in the health sector, all females.

A multiple Logistic regression model fitted considering the test result (positive vs. negative) as a dependent variable found that the presence of respiratory symptoms (OR 8.51; 95% CI 5.21–13.88; *p* < 0.001) and working in the catering/food sector (OR 0.22; 95% CI 0.10–0.48; *p* < 0.001) were independent predictors for SARS-CoV-2 rapid test positivity (Table 3). The estimated odds ratio favored an increase of nearly 22% for test positivity in occupations related to the catering/food sector. There was no multicollinearity among the independent variables.

No significant differences in the likelihood of being SARS-CoV-2 positive were found for other job categories, smoking, patient contacts, history of travel, previous testing, medical history at univariate analyses; thus, they were not used as independent variables in regression analysis.

### 3.2. Results from the Second COVID-19 Screening Program One Month after Lockdown

Four hundred and sixty-two people (48% males) participated in the second two-day COVID-19 screening program (30th November–1st December 2020) conducted one month after lockdown initiation. The population was significantly older compared to the pre-lockdown screening population, with a mean age of 48 ± 17 years [min: 5, max: 90 years]. Seventy percent of the participants were tested for the first time. Eighteen percent of the total population had already attended the pre-lockdown COVID-19 screening program.

As expected, the freedom of movement restrictions showed a markedly lower screening participation rate to the second program. However, there was no significant difference in the proportions of participants in most job categories between the first and second screening programs. Only the proportion of those who worked in the education sector halved in the second screening program (15 vs. 35% within the whole population).

Twenty-two out of four hundred and sixty-two (4.7%) participants had positive rapid antigen test results, with a subsequent RT-PCR confirmation. The prevalence of new infections decreased by 50% (4.7% from 8%) at the time point of the lockdown. The newly diagnosed participants belonged to the age group 19 to 69 years. The most considerable reduction in the prevalence of newly diagnosed cases (−94%) was observed in men aged 19–39 years (9% from 35% of the infected participants). Conversely, a 62% increase was observed in the prevalence of new cases for women aged 19–39, which constituted one-quarter of the infected participants at the time point of the lockdown.

Similar to previous results, one-third of the newly diagnosed patients were asymptomatic. One-third of the patients reported intra-family transmission, while the remaining two-thirds were “orphan” coronavirus cases. The distribution of positive and negative test results stratified by job group is presented in Table 4.

Positive participants were less likely to be retirees (5% vs. 23%, 0.004) than negative participants. Conversely, positive participants were more likely to be unemployed (39% vs. 12%, *p* = 0.020) than negative participants.

The effectiveness of infection control measures according to the type of job is presented in Table 5.

We found that the percentage of positive participants who worked in the education sector was significantly reduced in the second screening program compared to the first one. On the contrary, the percentage of unemployed infected individuals was significantly higher in the second screening program than in the first one (Table 5).

A multiple logistic regression model was conducted considering the test result (positive vs. negative) as a dependent variable based on the second screening program (Table 6). The analysis showed that the presence of respiratory symptoms (OR 8.61; 95% CI 2.83–26.15; *p* < 0.001) and the average weekly supermarket visits (supermarket visit frequency) during the last month (OR 1.27; 95% CI 1.12–1.45. *p* < 0.001) were independent predictors for SARS-CoV-2 rapid test positivity. No significant differences in the likelihood of being SARS-CoV-2 positive were found for various job categories, patient contacts, history of travel, previous testing, medical history, and smoking at univariate analyses; thus, they were not used as independent variables in the regression analysis.

## 4. Discussion

For the first time, we evaluated the changes in proportions of rapid antigen test positivity from the perspective of workers’ vulnerability through two passive surveillance programs conducted at two exceptional time points, before and one month after lockdown, and evaluated if the changes to positivity rates were significant across various job groups. This study showed that positive participants were more likely to work in the catering/food sector at the pre-lockdown time point than negative participants. The estimated odds ratio favored an increase of nearly 22% for test positivity in occupations related to the catering/food sector. A nearly 50% decrease in new COVID-19 cases was detected one month after lockdown restrictions, while there were no significant differences in the likelihood of being SARS-CoV-2 positive for different job categories or other sociodemographic characteristics. Interestingly, the supermarket visit frequency was predictive of positivity after COVID-19 restrictions. Finally, the presence of respiratory symptoms was a stable, independent predictor for SARS-CoV-2 rapid test positivity both at pre-lockdown and lockdown periods. However, one-third of the newly diagnosed patients were asymptomatic at both screening programs.

Interestingly, we found that employees in the catering/food sector experienced higher odds of COVID-19 positivity than those employed in other job categories. In accordance with the present results, the food production/processing sector has been identified by European Centre for Disease Prevention and Control (ECDC) as a potential hotspot for COVID-19 clusters or outbreaks [10]. Several distinct job characteristics could expose individuals to a high risk of contracting COVID-19, such as exposure to the virus due to the proximity to others, face-to-face discussions, and interactions with external customers or the public [11,12,13,14]. While there are certainly employers in the food/catering industry who provide high-quality jobs, by and large, the sector consists of very low-wage jobs with few benefits, and many restaurant workers live in poverty or near-poverty [15,16,17]. Moreover, the catering/food sector has traditionally been the sector with the highest percentage of foreign workers with different cultural and social backgrounds. The existing body of research supports a relationship between social class and health and found that substantive health disparities exist between different occupational statuses, as occupation is undoubtedly the bedrock for class differentiation in modern societies [18]. It has been recently documented that low-status workers are less likely to demand risk reduction equipment and infection control measures or have the bargaining power to obtain it [12]. Low-status workers are also less likely to be perceived as valuable and hard to replace by their employers, understand COVID-19 transmission routes, and comply with risk reduction strategies or implement their own [12].

Although a higher risk of in-hospital transmission in healthcare workers was expected, in this study, we found that healthcare workers did not experience higher odds of positivity than the rest of the employees. Studies have demonstrated that frontline health care workers have a 3.4-fold greater risk of infection with COVID-19 than those in the general population [19]. However, despite the high risks often faced by physicians, nurses, and other health personnel, workers in lower status jobs are generally more likely to be exposed to COVID-19 in the workplace than those in higher status occupations, as previously mentioned [19]. Higher-status workers have better access to risk mitigation measures, such as frequent sanitation, enforced distancing, personal protective equipment, and better ventilating and air filtration systems [12,13,14].

The Office for National Statistics reported that the working-age population (aged 20 to 64 years) suffered high levels of COVID-19 mortality [20]. However, there are little data supporting differences in infection rates amongst different occupations [21]. A work-related vulnerability to COVID-19 and other infections has been reported in occupations with daily prolonged, proximal contact with people [22]. In that context, apart from the food production/processing sector, the education, childcare sectors, sales, and retail sectors, bus/coach/taxi drivers and construction workers have also been proposed as occupations with a high potential for exposure and outbreaks. Another study, performed from June to October 2020 in the Netherlands, found that hospitality and public transport workers, driving instructors, hairdressers, and aestheticians had higher test positivity compared with a reference group of individuals without a close-contact occupation [11]. However, in this study, certain occupations with high SARS-CoV-2 infection risk, such as food processing, were not identified, given the limited number of occupational categories in the questionnaire [11].

Moreover, a gendered impact of the COVID-19 has been recognized. Women were overrepresented in certain healthcare occupations. In addition, women are disproportionately employed in higher COVID-19 risk occupations, such as cleaners and personal care workers, including long-term-care personnel. These work categories often have low pay and hazardous conditions and require direct contact with individuals [23].

Accordingly, we found that all the newly detected patients who worked in the health sector were females (5% within the positive) at the pre-lockdown time point. In a separate analysis, we did not detect any gender-related differences in the prevalence of positive tests among occupations before and during the lockdown.

Another important finding of this study was that one-third of the newly diagnosed patients belong to the teaching and education sector before lockdown, 40% of those were asymptomatic, and 20% of the newly detected patients were university students. There is limited information on the role of educational settings in COVID-19 spread and the extent to which children, adolescents, and university students may contribute to overall transmission. The balance of evidence from a large targeted population and school studies so far suggests that children (especially younger children) are less susceptible to viral infection than adults. Generally, SARS-CoV-2 infections and outbreaks were uncommon in educational settings. The investigations of cases identified in these settings suggest that in-school child to child transmission is not the primary cause of SARS-CoV-2 infection in children, particularly in preschools and primary schools [24]. In that context, the importance of controlling community transmission to protect educational settings has been emphasized [25].

Contradictory data are also present. A survey of more than 1900 American colleges and universities has revealed there were more than 397,000 cases and at least 90 deaths since the pandemic began, according to the New York Times coronavirus tracker [26]. Our results are consistent with these studies highlighting that young people and children may be important sources of asymptomatic transmission [27,28,29,30]. In the emerging COVID-19 context, schools and universities have implemented several measures to slow the spread of the virus. We found that school and university closures due to the COVID-19 lockdown had a clear impact on epidemic dynamics as the percentage of positive participants who belonged to the education sector was significantly reduced in the second screening program compared to the first one. We found that the implementation of infection control measures reduced by almost half the prevalence of COVID-19 in the screened population within a month, suggesting that lockdown measures constituted a strategy to prevent workers from becoming colonized or infected by SARS-CoV-2 rapidly. Lockdowns are the primary measures for the control of the virus and served as a proxy for whether workplaces and workers employ effective COVID-19-related risk reduction strategies. Despite the widely-recognized drastic reduction in both the quality and quantity of working, as well as the increased fear and stigmatization of exposure to COVID-19 at workplaces, there are limited data regarding the impact of the restrictive lockdown measures on infection rates amongst different occupations [31] that can mitigate the negative consequences, however. An analysis of deaths involving COVID-19 in different occupational groups aged 20 to 64 years in England and Wales showed that among health- and social-care professionals, the COVID-19-related death rates in men were around three times higher when they were infected before lockdown than during lockdown. Similarly, in women, rates were around two times higher [32]. However, the same analysis showed that the rates of COVID-19-related deaths were significantly lower in all occupation groups during lockdown when compared with the rates where the infection was likely to have been acquired before lockdown.

This study showed no significant differences in the likelihood of being SARS-CoV-2 positive for different job categories during the lockdown, an expected outcome given the paucity of jobs following the initiation of widespread restrictions.

Importantly, an insignificant percentage (approximately 10%) of the study population in both screening programs were unemployed. Importantly, the percentage of unemployed infected individuals was significantly higher in the second screening program than in the first one. Therefore, equally important for controlling COVID-19 are effective community-based and free or low-cost testing and tracing programs centered on the uninsured and underinsured population who had difficulty obtaining medical care.

Interestingly, we found that during the lockdown period, supermarket visit frequency constituted an independent predictor of COVID-19 positivity. This finding is suggestive of viral transmission in supermarkets frequented by many people during the lockdown. Although the high COVID-19 incidence at the municipal level was expected to be associated with a slowdown in sales due to the increased fears of being infected, it seems that supermarkets and online food retailers were crowned as COVID-19 winners during the lockdowns [33]. Greater numbers of people in stores and queues increase the likelihood of infection and make it challenging to ensure that social distancing restrictions are maintained, acting as a route for the spread for both clients and workers [34,35]. The evidence from this study suggests that close monitoring is essential for preventing the large-scale spread of the virus in such places.

COVID-19 is a disease of the pulmonary system, presenting with fever, cough, and shortness of breath, as well as anosmia and ageusia [36]. Multiple logistic regression analyses revealed that participants with respiratory symptoms were eight times as likely to screen positive for the coronavirus disease at both screening time points. On the other hand, another important finding was that one-third of the positive participants were asymptomatic in both screening programs. Our findings accord with earlier studies, which supported that asymptomatic infection rates range globally between 18% to 42% in different populations. There is evidence to support that a large proportion of positive employees self-reported as asymptomatic [37]. A conclusion drawn from the present study is that to increase the effectiveness of screening programs, people should be tested whether or not they exhibit any signs or symptoms of the disease.

Despite the insights provided by this study, our analysis has several important limitations. First, although our study tried to give a picture of the local impact of COVID-19 in the municipality of Volos, we do recognize that it was limited by the small sample size. Even though we aimed to have a larger sample size, the actual participation rate was much lower and may be subject to selection bias, though it is difficult to evaluate if the bias would favor higher or lower rates of participation among those likely to be positive. The free movement restrictions, the stigma of a positive COVID-19 test, and the fear of infection emerged strongly as the most significant barriers to screening. Second, all data, except for rapid antigen tests and RT-PCR results, were self-reported and may be subject to recall bias. Moreover, considering the specificity and sensitivity of the applied rapid antigen test, the expert manipulation and the clean vision of the collected samples, and that RT-PCR confirmed all positive cases, it cannot be determined whether there were lateral flow inhibitors in samples that could lead to false-negative rapid test results in specific/symptomatic cases or not. Furthermore, the lateral flow immunoassay color intensity is load-dependent, and full-colored and semi-colored were considered positive tests according to the kit protocol. However, it is unknown if there were any samples with no viral load from an infected individual to have false-negative results. Moreover, data on jobs used here may not provide an accurate picture of employment or occupational distributions during the pandemic. Our asymptomatic rate estimates may be inflated because of the 15-day time window imposed in the questionnaire. Therefore, some individuals reporting no symptoms may have been symptomatic before the 15-day window.

However, this is the first study assessing the risk of SARS-CoV-2 positivity according to the type of essential services and evaluating the direct impact of containment measures at a local level in Greece. Furthermore, in awareness of the priorities of the scientific community and Food and Drug Administration (FDA), to design a test with the best sensitivity for the virus detection, and considering current epidemiological aspects for SARS-COV-2, this is the first completed attempt that targets the detection of the virus itself in a community-based population sample, through an in toto repeated antigen testing strategy, from the angle of workers’ vulnerability to coronavirus. We did not only detect the virus in individual specimens through molecule affinity but also performed a surveillance regimen for COVID-19 in the community.

## 5. Conclusions

This project was the first comprehensive investigation of the prevalence of SARS-CoV-2 infection using rapid antigen testing and the impact of the lockdown measures in the proportion of positive results from the angle of workers’ vulnerability to COVID-19 in Greece. Positive participants were more likely to work in the catering/food sector at the pre-lockdown evaluation compared to negative participants. Lockdown restrictions halved new COVID-19 cases. No significant differences in the likelihood of being SARS-CoV-2 positive for different job categories or other sociodemographic characteristics were detected during the lockdown. Interestingly, supermarket visits were predictive of positivity after COVID-19 restrictions. Finally, the presence of respiratory symptoms was a stable, independent predictor for SARS-CoV-2 rapid test positivity both at the pre-lockdown and lockdown periods. Nevertheless, testing should be applied to a broad population because of the potential for asymptomatic disease.

The main contribution of this study has been to confirm that offering repeated community-based interventions with low-cost repeated antigen testing regimens that detect the virus itself in a population-based sample regardless of symptoms promptly is a crucial parameter to identify the potential modes of transmission, reveal potential lockdown factors that have a direct impact on the contagion level, effectively evaluate the impact of infection control strategies, and reduce epidemic spread.

## Figures and Tables

**Table 1 ijerph-18-01638-t001:** Demographic characteristics and positivity for SARS-CoV-2 of the study population according to the type of job.

Type of Job	N (% of the Whole Sample, *N* = 1054)	N of Males (% within the Group)	Mean Age (Years ± SD)	N of Positive Participants (% of within the Group)
Education sector				
Total	329 (31)	166 (50)	27.3 ± 4.8	30 (9)
Students	93 (9)	52 (56)	14.0 ± 2.5	7 (8)
University students	172 (16)	93 (54)	21.0 ± 2.0	18 (10)
Teachers	64 (6)	21 (33)	46.3 ± 9.8	5 (8)
Retirees	190 (18)	100 (53)	67.0 ± 8.1	7 (4)
Civil servants	142 (14)	58 (41)	48.2 ± 8.2	6 (4)
Self-employed	110 (10)	63 (57)	43.4 ± 11.0	8 (7)
Unemployed	108 (10)	25 (23)	43.9 ± 14.4	8 (7)
Private employees	65 (6)	38 (59)	39.9 ± 11.7	7 (11)
Catering/food sector	48 (5)	34 (71)	38.7 ± 12.6	17 (35)
Health sector *	31 (3)	7 (22)	42.4 ± 11.6	4 (13)
Army forces	31 (3)	29 (94)	36.3 ± 10.8	1 (3)

* Health sector included physicians, nurses, and physiotherapists.

**Table 2 ijerph-18-01638-t002:** Distribution of positive and negative test results by job type, Pre-lockdown screening program (N = 1054).

Type of Job	Total *N* = 1054	N of Negative Participants (% of within the Negative, *n* = 966)	N of Positive Participants (% of within the Positive *n* = 88)	*p*-Value ^1^
Education sector				
Total	329 (31)	299 (31)	30 (34)	NS
Students	93 (9)	86 (9)	7 (8)	NS
University students	172 (16)	154 (16)	18 (20)	NS
Teachers	64 (6)	59 (6)	5 (6)	NS
Retirees	190 (18)	183 (19)	7 (8)	0.005
Civil servants	142 (14)	136 (14)	6 (7)	0.036
Self-employed	110 (10)	102 (11)	8 (9)	NS
Unemployed	108 (10)	100 (10)	8 (9)	NS
Private employees	65 (6)	58 (6)	7 (8)	NS
Catering/food sector	48 (5)	31 (3)	17 (19)	<0.001
Health sector	31 (3)	27 (3)	4 (5)	NS
Army forces	31 (3)	30 (3)	1 (1)	NS

Notes: Data are expressed as frequency (percentage). ^1^ Comparisons between negative and positive participants, by job group.

**Table 3 ijerph-18-01638-t003:** Multiple logistic regression model with test result (positive vs. negative) as a dependent variable.

Variables	B	S.E.	Wald	Sig	Exp(B)	95% CI for EXP(B)	
						Lower	Upper
Age	−0.002	0.009	0.052	0.819	0.998	0.981	1.016
Gender (Male Ref)	−0.163	0.245	0.445	0.505	0.849	0.525	1.372
Presence of respiratory symptoms	2.141	0.250	73.413		8.509	5.214	13.886
Catering/food sector (yes Ref)	−1.503	0.400	14.100		0.222	0.102	0.487
Retirees (yes Ref)	0.571	0.515	1.233	0.267	1.771	0.646	4.857
Civil servants (yes Ref)	0.578	0.473	1.495	0.221	1.782	0.706	4.499

**Table 4 ijerph-18-01638-t004:** Distribution of positive and negative test results by job type, Second screening program (N = 462).

Type of Job	Total (% of within the Study Population, *N* = 462)	N of Negative Participants (% of within the Negative, *n* = 440)	N of Positive Participants (% of within the Positive, *n* = 22)	*p*-Value
Education sector				
Total	*73 (16)*	70 (16)	3 (14)	0.837
Students	13 (3)	13 (3)	0	0.714
University students	46 (10)	44 (10)	2 (9)	0.904
Teachers	14 (3)	13 (3)	1 (5)	0.887
Retirees	102 (22)	101 (23)	1 (5)	0.004
Civil servants	69 (15)	67 (15)	2 (9)	0.638
Self-employed	55 (12)	53 (12)	2 (9)	0.821
Unemployed	60 (13)	51 (12)	9 (39)	0.02
Private employees	37 (8)	35 (8)	2 (9)	0.823
Catering/food sector	34 (7)	33 (7)	1 (5)	0.866
Health sector	14 (3)	13 (3)	1 (5)	0.775
Army forces	18 (4)	17 (4)	1 (5)	0.824

**Table 5 ijerph-18-01638-t005:** Comparisons of the proportion of positive participants by job type between the first (pre-lockdown) and the second (after one month of lockdown) screening programs.

Type of Job	First screening Program N of Positive Participants (% of within the Positive, *n* = 88)	Second Screening Program N of Positive Participants (% of within the Positive, *n* = 22)	*p*-Value
Education sector			
Total	30 (35)	3 (14)	0.014
Students	7 (8)	0 (0)	0.196
University students	18 (21)	2 (9)	0.173
Teachers	5 (6)	1 (5)	0.651
Retirees	7 (8)	1 (5)	0.492
Civil servants	6 (7)	2 (9)	0.508
Self-employed	8 (9)	2 (9)	0.675
Unemployed	8 (9)	9 (39)	0.025
Private employees	7 (8)	2 (9)	0.578
Catering/food sector	17 (18)	1 (5)	0.174
Health sector	4 (5)	1 (5)	0.735
Army forces	1 (1)	1 (5)	0.364

**Table 6 ijerph-18-01638-t006:** Multiple logistic regression model with test result (positive vs. negative) as a dependent variable.

Variables	B	S.E.	Wald	Sig	Exp(B)	95% CI for EXP(B)	
						Lower	Upper
Age ^1^	0.001	0.020	0.001	0.981	1.000	0.961	1.040
Gender (Male Ref)	1.228	0.714	2.953	0.086	3.413	0.841	13.845
Presence of respiratory symptoms	2.153	0.567	14.421		8.610	2.834	26.154
Average weekly supermarket visits ^1^	0.242	0.067	13.078	<0.001	1.274	1.117	1.453

^1^ Continuous variable.

## Data Availability

The data that support the findings of this study are available on request from the corresponding author, O.S.K. The data are not publicly available due to restrictions; they contain information that could compromise the privacy of research participants.

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
