# Peer review of "Repeated Antigen-Based Rapid Diagnostic Testing for Estimating the Coronavirus Disease 2019 Prevalence from the Perspective of the Workers’ Vulnerability before and during the Lockdown"

_ijerph, 2021, doi:10.3390/ijerph18041638_

Round 1
Reviewer 1 Report
Materials and methods:
There are no paragraph concerning PCR tests for SARS-COV-2; Serological test were described with many details (specifity, sensitivity, clinical accuracy, repeatability) – describe this for PCR test too , as still PCR reaction requires different conditions for different mRNAs please describe conditions of PCR reaction.
Authors performed statistical analysis but there are no information about tests and methods used – please add this paragraph to a proper section.
Results:
In tables 1, 4,5 there is a “catering sector” category in table 2: “food sector” – please use the same name for category i.e. “catering/food sector”; as I assume date concerning the same group (type) of workers;
In table 3 there is adnotation about asterix in the table – but there is no one; remove the adnotation;
Results and Discussion:
In table 2 there is a results of positive individuals in student’s group, which is higher (20%) than in group of workers of food sector (19%), but there are no description in results section and no discussion of this result in discussion section – please add appropriate paragraphs to both sections.
Author Response
COMMENTS FROM REVIEWER 1
- Materials and methods: There are no paragraph concerning PCR tests for SARS-COV-2; Serological test were described with many details (specifity, sensitivity, clinical accuracy, repeatability) – describe this for PCR test too, as still PCR reaction requires different conditions for different mRNAs please describe conditions of PCR reaction.
RESPONSE: We appreciate you taking the time to offer us your comments and insights related to the manuscript. We are appreciative of this valuable remark. In the revision, we introduce a paragraph concerning PCR tests for SARS-COV-2 (found on page 3, lines 103-111.
- Authors performed statistical analysis but there are no information about tests and methods used – please add this paragraph to a proper section.
RESPONSE: Thank you very much for this comment. In the revision, we provide information about tests and methods used (page 3, lines 112-122).
- Results: In tables 1, 4,5 there is a “catering sector” category in table 2: “food sector” – please use the same name for category i.e. “catering/food sector”; as I assume date concerning the same group (type) of workers;
RESPONSE: Thank you very much for this point. In the revision, we have written this term uniformly.
- In table 3 there is adnotation about asterix in the table – but there is no one; remove the adnotation;
RESPONSE: The annotation in table 3 specifies that age is a continuous variable. However, per your suggestion, we removed it.
- Results and Discussion: In table 2 there is a results of positive individuals in student’s group, which is higher (20%) than in group of workers of food sector (19%), but there are no description in results section and no discussion of this result in discussion section – please add appropriate paragraphs to both sections.
RESPONSE: We are appreciative of this valuable remark. We really agree that this addition will improve the quality of our manuscript. In the revision, we presented this finding in the Results Section (page 4, lines 157-160) and discussed this issue in the Discussion Section on pages 8-9, lines 299-321.
We appreciate all of your insightful comments. We found them quite useful as we approached our revision. We are grateful for the time and energy you expended on our behalf.

Reviewer 2 Report
Although this work shows clear findings about differences in COVID-19 infections rate amongst different occupation (before lockdown) and differences before and after lockdown, I have a few concerns which could be improved:
- In the Methods, the company and country producing the Viva Diag kit should be stated.
- In the Methods, the authors should give the details of the RT-PCR.
- In the Results, please define in more details about low- and high-status workers
- In the Discussion, authors should discuss more about COVID19 cases in other countries or about other viral infections, in the context of differences in infections rates amongst different occupations or before/after lockdown.
- Many sentences are not referenced (there are only 11 references)
Author Response
COMMENTS FROM REVIEWER 2
- Although this work shows clear findings about differences in COVID-19 infections rate amongst different occupation (before lockdown) and differences before and after lockdown, I have a few concerns which could be improved:
RESPONSE: Thank you very much for your kind words about our paper. We are delighted to receive positive feedback from you. We appreciate you taking the time to offer us your comments and insights related to the manuscript. The thoughtful guidance provided by you has helped improve the quality of our manuscript. In the following pages are our point-by-point responses to each of your comments.
- In the Methods, the company and country producing the Viva Diag kit should be stated.
RESPONSE: Thank you for this point. In the revision, the company and country producing the Viva Diag kit are stated (on page 2, line 91).
- In the Methods, the authors should give the details of the RT-PCR.
RESPONSE: Thank you for this interesting remark. In the revision, we give the details of the RT-PCR on page 1, lines 103-111.
- In the Results, please define in more details about low-and high-status workers
RESPONSE: Thank you for this direction. In the revision, we deal with this issue in the discussion section on pages 7-8, lines 252-275.
- In the Discussion, authors should discuss more about COVID19 cases in other countries or about other viral infections, in the context of differences in infections rates amongst different occupations or before/after lockdown.
RESPONSE: Thank you for this valuable suggestion. We really agree that these additions will improve the quality of our manuscript. In the revision, we tried to deal with these issues on page 8, lines 276-298 and page 9, lines 336-347.
- Many sentences are not referenced (there are only 11 references)
RESPONSE: Thank you for this comment. We revised the manuscript accordingly and the appropriate references have been added.
We found your feedback very constructive. We tried to be responsive to your concerns. We really thank you for taking the time and energy to help us improve this paper.

Reviewer 3 Report
Dear Authors,
You have written an interesting study. However, there are some small parts that need to be addressed.
Materials and Methods section should be amended.
-What was the sample size (you just reported the population of the City and nod how many people you tested) report
-how many tests were inconclusive / report
Overall in my opinion the paper is to be accepted after a minor revision.
Kind regards
Author Response
COMMENTS FROM REVIEWER 3
- Dear Authors, You have written an interesting study. However, there are some small parts that need to be addressed.
RESPONSE: Thank you very much for your kind words about our paper. We appreciate you taking the time to offer us your comments and insights related to the manuscript.
- Materials and Methods section should be amended.
RESPONSE: Thank you for this comment. In the revision, the Materials and Methods section was revised. More specifically, we have introduced a paragraph concerning PCR tests for SARS-COV-2 (found on page 3, lines 102-111) and we provided information about tests and methods used (page 3, lines 112-122). We hope you find these revisions rise to your expectations.
- -What was the sample size (you just reported the population of the City and nod how many people you tested) report-how many tests were inconclusive / report
RESPONSE: Thank you for this comment. The sample size of this study is discussed when results are presented (Results Section: page 3, line 122 “1054 volunteers participated in the pre-lockdown screening program” and page 5, line 172 “462 people participated in the second two-day COVID-19 screening program conducted one month after lockdown initiation”. However, per your suggestion, we provide information regarding the sample size of this study in an opening paragraph of this section (page 4, lines 124-128). No inconclusive results were found. Nevertheless, according to World Health Organizations’ recommendations, all cases manifesting a rapid test positivity for SARS-CoV-2 were immediately resampled for NAAT only for SARS-CoV-2 confirmation.
- Overall in my opinion the paper is to be accepted after a minor revision.
RESPONSE: We are delighted to receive positive feedback from you. We are grateful for the time and energy you expended on our behalf.
